# Experimental Analysis of Lightweight Fire-Rated Board on Fire Resistance, Mechanical, and Acoustic Properties

**Ming Chian Yew** [1,*] **, Ming Kun Yew** [2] **and Richard Kwok Kit Yuen** [3]

1   Department of Mechanical and Material Engineering, Lee Kong Chian Faculty of Engineering and Science, Universiti Tunku Abdul Rahman, Cheras, Kajang 43000, Malaysia
2   Department of Civil Engineering, Lee Kong Chian Faculty of Engineering and Science, Universiti Tunku Abdul Rahman, Cheras, Kajang 43000, Malaysia; yewmk@utar.edu.my
3   Department of Architecture and Civil Engineering, City University of Hong Kong, Hong Kong 999077, China; richard.yuen@cityu.edu.hk
*   Correspondence: yewmc@utar.edu.my or yewmingchian@gmail.com; Tel.: +60-390860288

**Abstract:** Using lightweight fire-rated board (LFRB) presents cost-effective opportunities for various passive fire protection measures. The aim of the project is to develop an LFRB with enhanced fire resistance, acoustic properties, and mechanical properties. These properties were determined using a Bunsen burner, furnace, energy-dispersive X-ray, impedance tube instrument, and Instron universal testing machine. To fabricate the LFRBs, vermiculite and perlite were blended with flame-retardant binders, and four types of LFRBs were produced. A fire test was conducted to compare the fire-resistance performance of the LFRBs with a commercially available flame-retardant board. The B2 prototype showed exceptional fire-resistant properties, with a temperature reduction of up to 73.0 °C, as compared to the commercially available fire-rated magnesium board. Incorporating nano chicken eggshell into the specially formulated flame-retardant binder preserved the LFRBs' structural integrity, enabling them to withstand fire for up to 120 min with an equilibrium temperature of 92.6 °C. This approach also provided an absorption coefficient of $\alpha$ = 2.0, a high flexural strength of 3.54 MPa, and effective flame-retardancy properties with a low oxygen/carbon ratio of 2.60. These results make the LFRBs valuable for passive fire protection applications in the construction and building materials industry.

**Keywords:** lightweight; fire-resistant board; intumescent; passive fire protection





## 1. Introduction

The implementation of active and passive fire prevention (APFP) systems is crucial in improving fire prevention and protection measures and safeguarding both lives and property in buildings [1]. Flame-retardant boards are a type of passive fire prevention (PFP) system that play a decisive role in modern-day building fire security protocols. Flame-retardant boards are considered to be beneficial during a fire incident because they can offer valuable time for evacuation and delay building collapse. In recent years, intumescent fire-protective materials have emerged as an effective barrier in various PFP systems [2–4]. In addition to fire-rated boards, fire doors, fire dampers, and firewalls are also considered to be vital elements of fire safety in modern buildings [5]. Unlike active fire prevention (AFP) systems, PFP systems do not require human intervention to work and can automatically prevent the spread of fire upon contact. Therefore, this experimental study focuses on the combustion protection materials principle to maintain the fire protection, sound insulation, and mechanical properties of lightweight flame-retardant boards (LFRBs) in building construction.

In practice, fire-retardant boards have several important functions. They can impede the spread of flames in structures and serve as a barrier to prevent fire hazards, offering a safe environment for occupants [6,7]. This study emphasizes the significance of LFRBs in

providing sufficient time for evacuation during a fire emergency and reducing the risks of fire incidents. In addition, the use of water-based intumescent binder (W-IB) has become increasingly popular in fire-retardant construction materials due to its benefits, such as being lightweight, environmentally friendly, and low-odor. Various studies have reported that intumescent materials (IM) have exhibited good physical and chemical properties and effective flammability prevention in wood and steel structures [8–11]. Intumescent material, commonly used in buildings and structures, is a fire-resistant coating that provides passive fire protection by delaying ignition and reducing the burning rate. When exposed to heat, IM rapidly expands and forms a dense, porous char layer with minimal thermal conductivity. This char layer operates as an insulating barrier, preventing the substrate from reaching high temperatures and causing structural instability. IM's low density and stability ensure that it can provide long-lasting protection against fires, making it a vital component of passive fire protection systems. IM is an effective solution for minimizing fire damage and preventing structural failure [10,11].

Intumescent material has the ability to function both as a thin film coating and a binding agent to improve combustion protection. Binders are often used in conjunction with fire-resistant boards since they are inflexible prefabricated materials, as reported by Ariyanayagam and Mahendran in 2017. Industrial fire-rated boards are typically made of materials such as cement, gypsum, and magnesium oxide, as evidenced by previous studies [12–15]. However, the existing commercial fire-resistant boards share common issues of being too heavy and high density [14,16]. The combination of IB and low-density flame-retardant materials presents a new and feasible approach to safeguarding substrates against reaching critical temperatures in the effect of flames. These fire safety measures are economical, lightweight, provide acoustic insulation, and can be customized to suit a broad spectrum of applications [17,18].

Based on the substantial potential of intumescent materials that has been repeatedly reported by researchers, the current focus of the community is on developing more reliable, efficient, and effective flame-retardant formulations [19–21]. The objective of this research project is to construct a fire-resistant board that includes an intumescent flame-retardant binder, which expands in a controlled manner when exposed to fire, creating an insulating carbonaceous char that safeguards the substrate from fire consequences. The board is designed to assist in subdividing buildings into smaller sections to impede or halt the spread of fire or smoke from one area to another. This novel and innovative approach is aimed at enhancing building safety by providing effective measures to control the spread of fires [22–25]. This makes the binder critical for building protection and can provide additional time for occupants to escape unharmed in a fire burst by blocking the flames and fumes and insulating the heat.

In addition, it is important to observe that most carbonate fire-retardant fillers, with the exception of magnesium and calcium carbonates (i.e., eggshell nano bio-filler), release non-combustible carbon dioxide ($CO_2$) at high temperatures to block the expansion of flames. However, calcium carbonates and magnesium release $CO_2$ under the temperature of 1000 °C, which can aid in the development of an insulating charred layer during the resulting breakdown of oxides [9,26].

IM is becoming increasingly popular as a flame-retardant material in construction engineering due to its numerous advantageous properties. Several in-depth investigations have revealed that IB exhibits impressive flammability properties [27–33]. However, fire-resistant boards that incorporate intumescent fire-protective binders in an appropriate manner have not been developed thus far. This research, therefore, focuses on the development of LFRBs with a density of approximately $600 \pm 50 \text{ kg/m}^3$ by blending W-IBs with vermiculite and perlite. The aim is to attain a 120 min fire rating for use in architecture, building, and construction applications.

## 2. Materials and Methods

To create a novel water-based intumescent binder with eco-friendly flame-retardant additives, the first phase of this research project involved mixing four specific formulations using a high-speed mixer capable of reaching up to 3000 rpm. The formulation included fire-retardant additives: APP, PER, MEL, and expandable graphite-EG. To achieve flame retardancy, a variety of flame-retardant fillers were employed, including magnesium hydroxide ($Mg(OH)_2$), aluminum hydroxide ($Al(OH)_3$), nano chicken eggshell, calcium silicate ($CaSiO_3$), and titanium dioxide ($TiO_2$). Finally, all the components were bound together with a water-based vinyl acetate copolymer (VAC) binder. The fire resistance of the intumescent binders coated on galvanized steel sheets were characterized through a Bunsen burner test and the temperature variation at every two minutes was recorded. Table 1 presents the different formulations of intumescent binders used in this study.

**Table 1.** Formulations of intumescent binders (A1–A4).

| Ingredients | The Weight Proportions Used in the Formulation of Intumescent Coatings Are Expressed as Parts by Weight (wt.%) | | | |
|---|---|---|---|---|
| | **A1** | **A2** | **A3** | **A4** |
| Flame-retardant additives | | | | |
| APP acts an acid source | 20 | 20 | 20 | 20 |
| PER acts a carbon source | 10 | 10 | 10 | 10 |
| MEL acts a blowing agent | 10 | 10 | 10 | 10 |
| EG | 2 | 2 | 2 | 2 |
| Polymer binder | | | | |
| Vinyl acetate (VA) copolymer emulsion | 48 | 48 | 48 | 48 |
| Pigment | | | | |
| Titanium dioxide | 3.0 | 3.0 | 3.0 | 3.0 |
| Flame-retardant fillers | | | | |
| Aluminum hydroxide | 3.5 | 3.5 | - | - |
| Magnesium hydroxide | - | - | 3.5 | 3.5 |
| Calcium silicate | 3.5 | - | 3.5 | - |
| Nano chicken eggshell | - | 3.5 | - | 3.5 |

$Al(OH)_3$ and $Mg(OH)_2$ are widely used as flame-retardant fillers in various industries. $Al(OH)_3$ is primarily attributed to its ability to release water vapor when exposed to heat, which helps to suppress flames and lower the temperature of the material. Similarly to $Al(OH)_3$, $Mg(OH)_2$ undergoes endothermic decomposition when exposed to heat. It releases water vapor ($H_2O$) and magnesium oxide (MgO) as byproducts. The released water vapor absorbs heat energy from the flame and the material, thus reducing the temperature and slowing down the combustion process. $Mg(OH)_2$ contributes to smoke suppression during a fire. The released water vapor helps to cool down the gases and particles generated during combustion, minimizing smoke production. Additionally, the fine particles of MgO formed act as nucleating agents, promoting the formation of carbonaceous char, which can further reduce smoke generation. In addition, $CaSiO_3$ forms a layer of calcium oxide (CaO) and silicon dioxide ($SiO_2$), commonly known as lime and silica, respectively. This layer acts as a protective barrier, shielding the underlying material from direct contact with the flame and reducing the supply of oxygen necessary for the combustion process.

Nano chicken eggshells are predominantly composed of calcium carbonate ($CaCO_3$), which is a naturally occurring mineral. $CaCO_3$ possesses an inherent ability to release carbon dioxide ($CO_2$) when exposed to high temperatures. This endothermic reaction

absorbs heat and dilutes the concentration of flammable gases, leading to flame inhibition and reducing the spread of fire.

$TiO_2$ acts as a pigment disperser, helping to distribute other pigments, fillers, and additives evenly throughout the coating formulation. This improves the coating's stability, consistency, and overall performance. In addition, the combination of $TiO_2$ and flame-retardant additives helps to achieve the desired fire-resistant properties. These additives can include halogenated compounds, phosphorous-based compounds, or intumescent systems, which can act as flame retardants in the presence of a heat source. It can promote the formation of a more robust char layer when exposed to fire, providing additional insulation and protection to the substrate.

The following step of this investigation was to create and manufacture a new type of fire-resistant board, which had dimensions of $300 \times 300 \times 30$ mm. The four sets of intumescent binders were incorporated with the same amount of vermiculite and perlite, respectively, as matrix composite in fabricating the flame-retardant board as shown in Table 2.

**Table 2.** The density of lightweight flame-retardant boards (B1–B4).

| Flame-Retardant Board (Dimensions: $300 \times 300 \times 40$-mm) | B1 | B2 | B3 | B4 |
|---|---|---|---|---|
| Dried weight (g) | 2253 | 2249 | 2250 | 2258 |
| Density ($kg/m^3$) | 625.93 | 624.72 | 624.99 | 627.22 |

The final phase of the research project involved a physical evaluation of the fire protection, mechanical strength, and sound insulation properties of the fire-resistant boards. The assessments of fire, three-point flexural, and acoustic properties were evaluated to measure their performances. The temperature rise test and fire durability test were performed to determine the heat transmission, substantial leakage, and integrity failure of the fire-retardant boards.

## 2.1. Flame Resistance Performance

### 2.1.1. Fire Protection Test

The experimental research was divided into two sections for the purpose of fire resistance testing. The first section involved measuring the temperature outlines of the coated W-IBs on galvanized steel plates, as illustrated in Figure 1. Specifically, the intumescent materials (IMs) were applied onto a galvanized steel sheet with a dry thickness of 1.5 mm. To monitor temperature changes, a thermocouple plate was affixed to the opposite side of the galvanized steel plate and linked to a digital handheld thermometer. The device is equipped to record the temperature outline of each formulation, the time in terms of mins for time intervals of 2 min for 60 min. The test involved exposing the sample to a Bunsen burner for a duration of 60 min at a temperature flame of 1000 °C while maintaining a distance of around 80 mm, and monitoring the temperature of the sample at two-minute intervals. The Bunsen burner was estimated to have a gas consumption rate of approximately 165 g/h [24], whereas the second part was to examine the 2 h LFRB prototypes by comparing it with a magnesium oxide (MgO) board under the comparable Bunsen burner test set up to measure the temperatures of points T1 and T2, as shown in Figure 2.

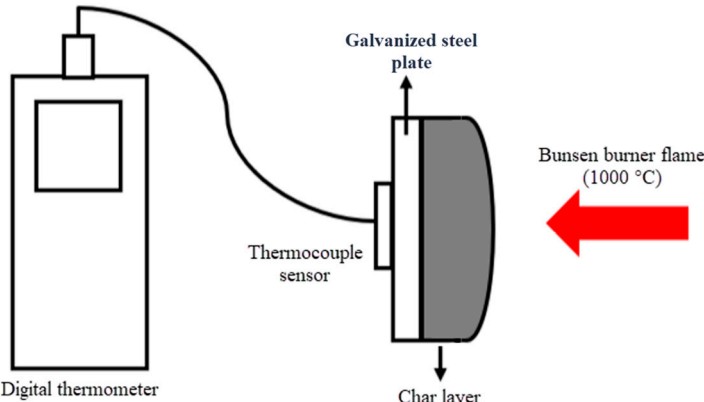

**Figure 1.** Bunsen burner test setup for intumescent binder (Part 1).

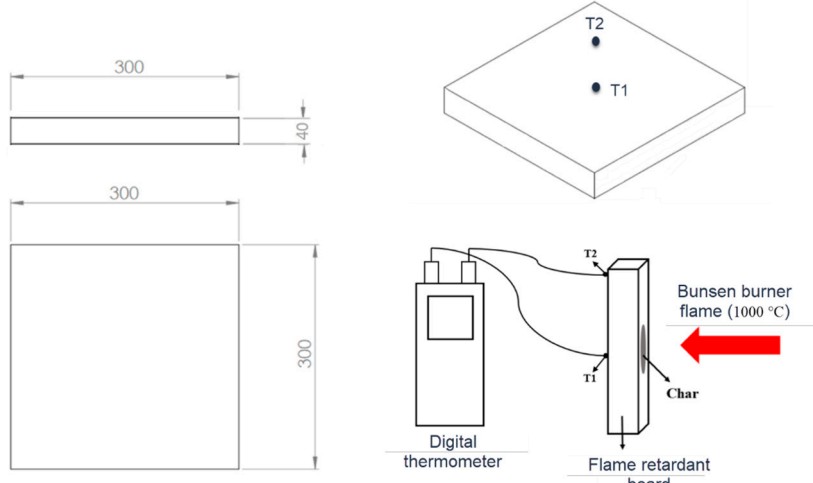

**Figure 2.** Dimensions of the fire-rated board (all in mm) and setup of fire test under 120 min of fire (Part 2).

### 2.1.2. Carbolite Furnace Test towards Char Strength

The point of conducting the furnace test is to evaluate the flame safety effectiveness of the IMs by observing the development of a char when exposed to temperatures of 500 °C and 600 °C, respectively, under a uniform temperature.

### 2.2. Impedance Tube Test

There are two tests involved in the impedance tube test which are the transmission loss test and the absorption properties test, as shown in Figure 3. Both the transmission loss test and the absorption test will be used together to obtain the acoustic insulation properties of the LFRBs.

VA-LAB4 software was used in processing the data obtained from the equipment which was the BSWA SW 422 Impedance Tube. The impedance tube test enables data to be obtained (sound absorption and transmission loss) in accordance with the ISO and ASTM standards. The LFRBs with three different samples size were used in this test which were 30, 60, and 78 mm in diameter. Firstly, a BSWA 1/4″ MPA416 microphone was inserted into a located slot in the impedance tube and connected to MC 3242 data acquisition hardware. In addition, the PA50 power amplifier was connected to the impedance tube, and the obtained data were utilized to operate the loudspeaker within the impedance tube.

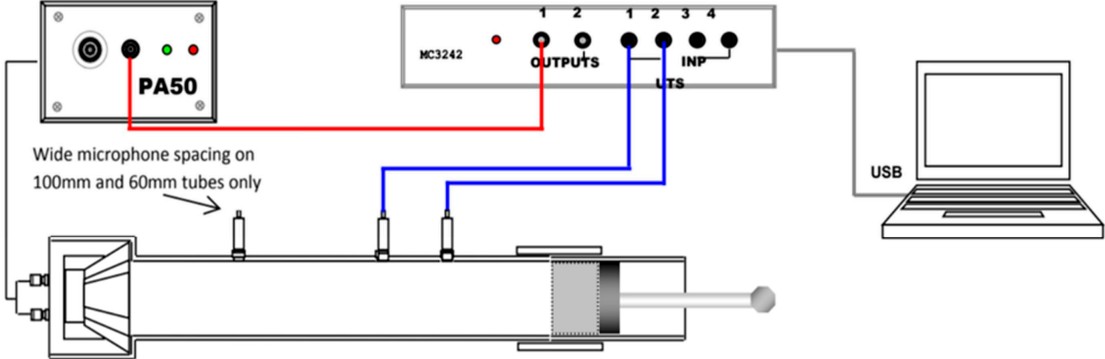

**Figure 3.** Absorption measurement setup for the flame-retardant boards.

First, the characteristic impedance, Z, was obtained as in Equation (1).

$$Z = \rho \cdot c \tag{1}$$

where
Z = characteristic impedance, $kg \cdot m^{-2} \cdot s^{-1}$;
$\rho$ = density of the medium, $kg/m^3$;
c = sound speed in the medium, m/s.

After obtaining the characteristic impedance, it was used and applied in Equation (2) in order to obtain the reflection coefficient (R).

$$R = (z_2 - z_1)/(z_2 + z_1) \tag{2}$$

where
R = reflection coefficient;
z = characteristic impedance, $kg \cdot m^{-2} \cdot s^{-1}$.

Next, the coefficient of reflection was substituted into Equation (3).

$$\alpha = 1 - |R|2 \tag{3}$$

$\alpha$ refers to the ratio of the amplitude of the reflected sound wave to the amplitude of the incident sound wave.

### 2.3. Three-Point Flexural Test

The specimens were created in accordance with ASTM D790 (ASTM, 2015) and had dimensions of $300 \times 30 \times 30$-mm. The prototypes were fixed to an Instron Micro Tester and subjected to compressive forces at a crosshead speed of 1 mm/min while maintaining a support span length of 150 mm. The samples were pulled apart until they broke or completely detached, as depicted in Figure 4.

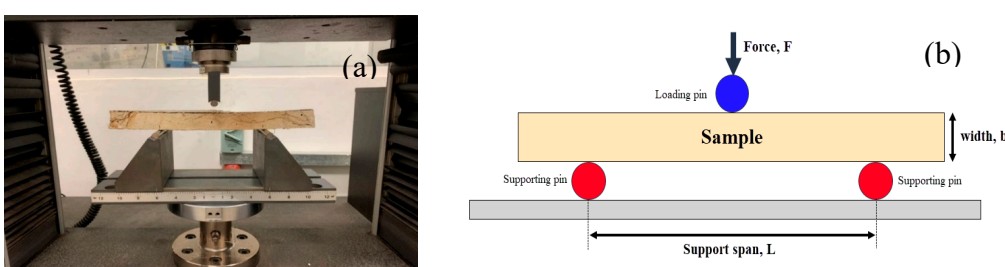

**Figure 4.** (**a**) Three-point flexural test setup. (**b**) Schematic diagram.

The flexural test is conducted to evaluate both the flexural strength, which is the maximum stress experienced by the sample's outermost fiber on the compression or tension side, and the flexural modulus, which is calculated from the slope of the stress versus the strain–deflection curve. These measurements are crucial in determining the sample's resistance to bending or flexure forces. The flexural stress and flexural modulus can be calculated using Equations (4) and (5), respectively.

$\sigma_f$ = the stress experienced by the outer fibers at the midpoint (MPa).

$\varepsilon_f$ = the deformation experienced by the outer surface expressed in mm/mm.

$E_f$ = the flexural modulus of elasticity, measured in MPa.

$F$ = the force applied at a specific point on the load–deflection curve, expressed in Newtons (N).

$L$ = the distance between the two supports, measured in millimeters (mm).

$b$ = the dimension of the test sample's width, measured in millimeters (mm).

$d$ = the measurement of the test sample's depth or thickness, expressed in millimeters (mm).

$D$ = the highest point of deformation experienced by the center of the sample, measured in millimeters (mm).

$m$ = the slope of the initial linear section of the load–deflection curve, expressed in Newtons per millimeter (N/mm).

Equation of flexural stress ($\sigma_f$):

$$\sigma_f(\text{MPa}) := \frac{3 \times \text{FL}}{2 \times \text{bd}^2}. \tag{4}$$

Equation of flexural modulus ($E_f$):

$$E_f(\text{MPa}) := \frac{\text{L}^3\text{m}}{4 \times \text{bd}^3} = \frac{\sigma_f}{\varepsilon_f}. \tag{5}$$

### 2.4. Energy Dispersive X-ray (EDX) Analysis

The EDX analysis is a technique that utilizes X-rays to determine the elemental composition of tested fire-retardant boards. This involves the release of X-rays when electrons move from higher to lower energy shells in an atom, with the energy of the X-rays being unique to the element and the transition involved. EDX analysis is useful for both qualitative and quantitative evaluations, allowing for the detection of elements and the determination of their concentrations in a sample, especially for carbon and oxygen.

## 3. Results and Discussion

### 3.1. Fire-Protective Performance

This fire test aims to analyze the reaction of the intumescent binder and the char formation. The temperature evolution between the galvanized steel plate coated with the binder and the uncoated plate during combustion were compared. This study evaluates the fire resistance properties of four different intumescent binders (A1–A4) through the Bunsen burner test.

During the Bunsen burner test, temperature measurement and recording were performed on the back side of the steel plate coated with various intumescent binders using a Type-K thermocouple sensor connected to a digital handheld thermometer. The measurements were taken continuously for a duration of 60 min. The maximum temperature of the unprotected galvanized steel plate attained about 542 °C after 60 min, while the maximum temperatures of the protected samples A1, A2, A3, and A4 were 198 °C, 178 °C, 257 °C, and 226 °C, respectively, as shown in Figure 5.

Figure 6 displays the thickness of the char layer in the samples coated with the IB formulation after the fire test. During the first 10 min of the test, the temperatures of the coated samples remained below 180 °C, demonstrating similar results. Thereafter, the temperatures of all the coated samples remained steady, indicating that the physical and chemical reactions of the IB formulation had concluded. In contrast, the temperature

of the galvanized steel plate, which was not protected, increased rapidly and attained a temperature of 508 °C after 10 min. The temperatures of samples A1–A4 reached a state of equilibrium after 30 min and remained nearly constant until the end of the test. Sample A2 had the best fire-protective performance, with an equilibrium temperature of 178 °C, which was significantly lower than the other samples.

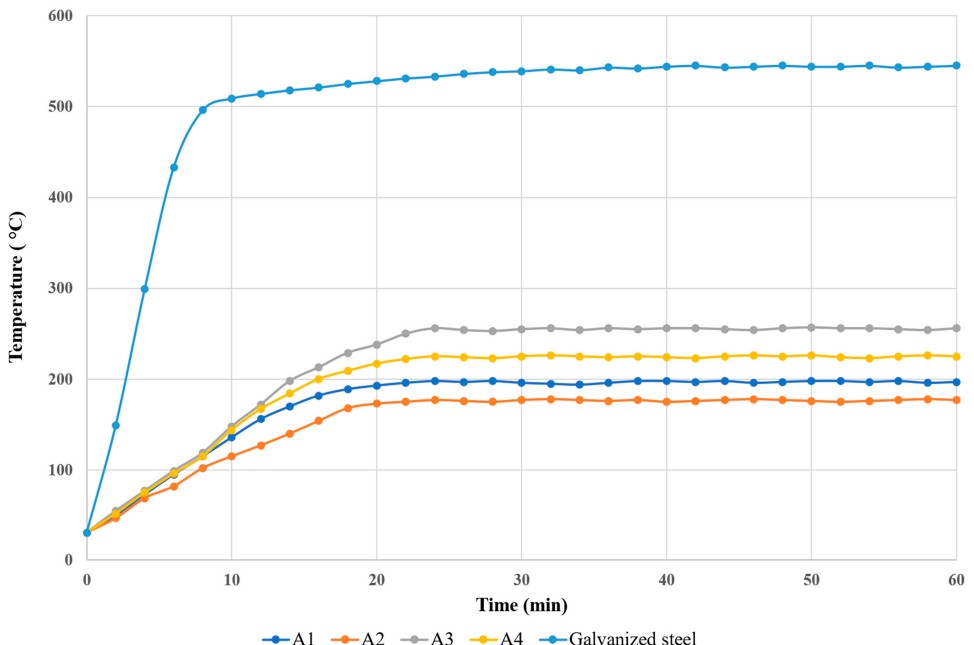

**Figure 5.** Evolution of temperatures on the protected and unprotected galvanized steel plates.

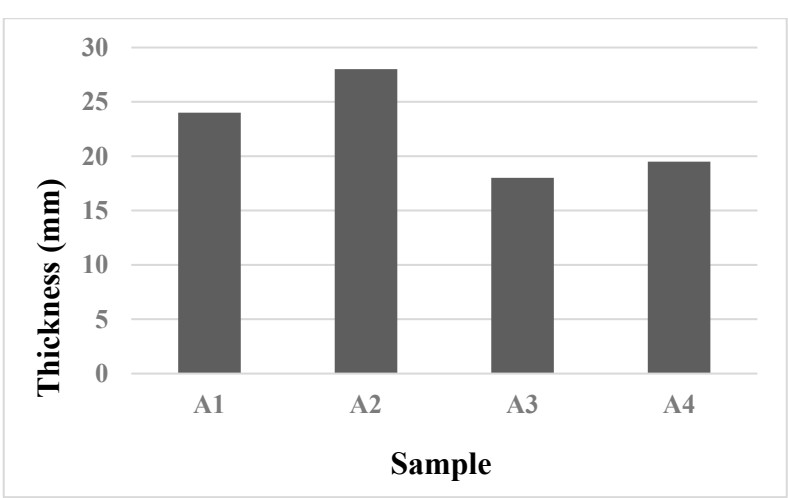

**Figure 6.** Char layer of samples A1–A4 after the Bunsen burner test.

The char layer formed on the coated samples showed positive fire-protective effects due to the presence of flame-retardant ingredients and nano CES at 3.5 wt.%, resulting in a thick and excellent char layer (approximately 28.0 mm) that acted as a barrier against fire while also facilitating the creation of a dense char structure. After 60 min of fire exposure, the equilibrium temperature for sample A3 was found to be the highest, reaching approximately 260 °C. This may be due to the utilization of flame-retardant fillers, such as $Mg(OH)_2$ and $CaSiO_3$, which caused minimal expansion of the char layer (measuring 17.0 mm) during the physical and chemical reactions between the binder and the flame-retardant ingredients, as compared to the other samples [24,30].

*3.2. Furnace Test*

The furnace test is conducted to evaluate the effectiveness of the fire protection provided by the coated samples when exposed to high temperatures of 500 °C and 600 °C, with a critical temperature of 400 °C, which is a significant temperature range for most house fires. The thickness of the char layer is measured to determine the samples' abilities to endure high temperatures and impede the spread of fire. Figure 7 shows the recorded thickness of the char layer. The results indicate that the addition of magnesium hydroxide in coating samples A3–A4 resulted in slightly thinner char layers when compared to coating samples A1 and A2. The reason for this phenomenon is believed to be the adverse impact of the flame-retardant filler, which impaired the fire protection capabilities due to limited char expansion, leading to insufficient insulation for shielding the primary layer of steel plate from heat.

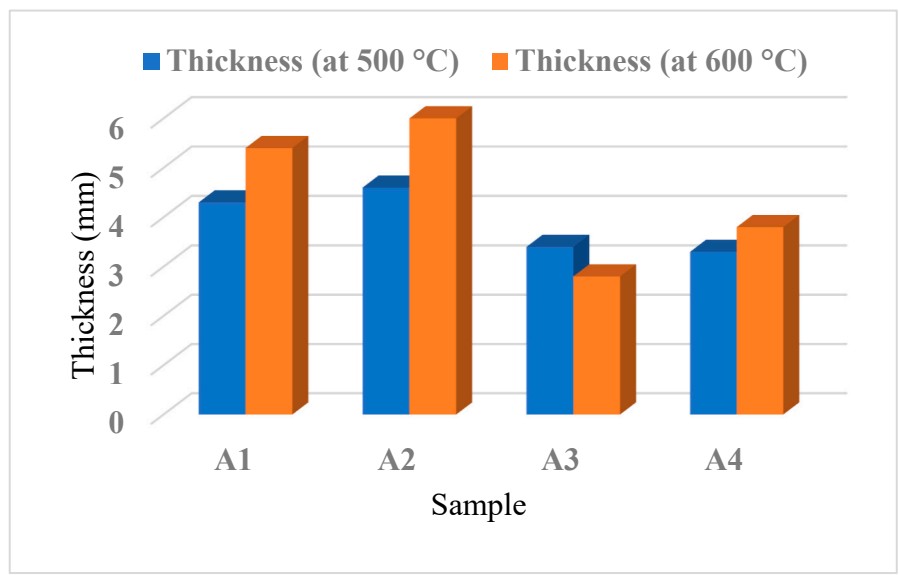

**Figure 7.** Char layer thicknesses of the coating samples measured at 500 °C and 600 °C.

In contrast, coating sample A3 exhibited different behaviors at the two tested temperatures. At 500 °C, the thickness of the char layer increased, likely due to the decomposition of $Mg(OH)_2$ which can envelop the flame and dilute flammable gases while excluding oxygen [34]. However, at 600 °C, the fire was able to propagate through the fire-protective barrier, resulting in a decrease in the thickness of the char layer for coating sample A3.

Sample coating A2 showed a better ability to expand the char layer at 500 °C and 600 °C compared to other samples, as evidenced by its highest char layer thickness values of 4.6 mm and 6.0 mm, respectively. This can be attributed to the presence of $Al(OH)_3$ and CES, which facilitated the expansion of the char layer through decarbonation during the burning process. The decarbonation process of CES resulted in the release of carbon dioxide, which was trapped in the residue and caused it to swell. On the other hand, coating sample A1 had a char layer thickness of 4.3 mm at 500 °C and 5.4 mm at 600 °C, with an expansion value of approximately 1.1 mm.

On the other hand, coating sample A1 exhibited a thicker char layer compared to samples A3 and A4. The addition of $Al(OH)_3$ to sample A1 might have facilitated strong reversibility of the dehydration reaction by combining the reactive surface of freshly formed alumina with water moisture inside the particle, resulting in a good flammability resistance filler and expansion of the char layer. This observation is in line with Packham's findings in 1996. Sample A4 showed an expansion in the thickness of the char layer by 3.3 mm at 500 °C and 3.8 mm at 600 °C, respectively. The combination of nano CES with $Al(OH)_3$ in sample A2 exhibited significant flame-retardant fillers; combining it with $Mg(OH)_2$ filler may have resulted in a poorer fire resistance performance. Therefore, it can be inferred that

the presence of Mg(OH)$_2$ filler may hinder the formation of a char layer and reduce the fire-protective performance.

### 3.3. EDX Test Analysis

In preparation for EDX analysis, all char samples were first coated with gold to avoid any charging effects and then subjected to a low beam energy of 1 kV to minimize the potential for thermal damage [27]. The results of the samples' content of oxygen, carbon, and phosphorus were tabulated in Table 3.

**Table 3.** The contents of oxygen, carbon, and phosphorus in the char samples.

| Samples | Oxygen | Carbon | Phosphorus | Ratio: Oxygen/Carbon |
|---------|--------|--------|------------|----------------------|
| A1 | 12.0% | 43.2% | 23.7% | 3.60 |
| A2 | 16.7% | 43.4% | 21.2% | 2.60 |
| A3 | 6.3% | 45.1% | 26.7% | 7.12 |
| A4 | 8.8% | 45.1% | 27.2% | 5.67 |

Additionally, coatings with lower oxygen-to-carbon ratios tend to produce thicker char layers due to the greater amount of carbon present. This increased carbon content in the char layer acts as an insulator, protecting the underlying steel from the high temperatures of the fire. Therefore, a lower oxygen-to-carbon ratio in an intumescent coating is generally considered desirable for achieving superior fire-protection performance [35].

The ratio of oxygen-to-carbon of samples A1 to A4 are presented in Table 3. Notably, the oxygen/carbon ratio of coating sample A2 is the lowest among the four samples. This lower oxygen/carbon content likely led to greater expansion of the char layer in sample A2 compared to the other samples.

It is widely acknowledged that flame-retardant materials containing phosphorus are highly effective. Phosphorus is a compound that operates by trapping radicals to impede flames in the gaseous phase, and by promoting the formation of carbon char in the condensed phase. The efficiency of phosphorus as a flame retardant relies on the material's chemical composition, as well as its ability to react with OH groups during a fire at elevated temperatures. Consequently, higher levels of phosphorus are more effective in intumescent binders that function as polymers.

In 2018, K. Md Nasir and colleagues conducted a study that found a positive correlation between the level of phosphorus in intumescent coating with calcium carbonate as the flame-retardant filler before and after the Bunsen burner test and EDX analysis [36]. On the other hand, the study also discovered that the use of intumescent coating with aluminum hydroxide fillers showed a negative impact on phosphorus content. However, aluminum hydroxide fillers were found to be effective in forming a char layer that can delay heat flow and the spread of combustible gas, resulting in good flame retardancy. The study also reported that coating sample A2 with aluminum hydroxide fillers had the lowest phosphorus content at 21.2%, while coating sample A4 with calcium carbonate fillers had the highest phosphorus content at 27.2%. These findings suggest that the condensed phase mechanism of phosphorus can be an effective means of flame retardancy in intumescent binders.

### 3.4. Testing for Fire-Rated Boards

Fire Endurance and Temperature Rise Tests

The purpose of this test is to assess the fire protection capabilities of the fire-resistant board prototypes (B1–B4). The temperature rise of each prototype was recorded over a period of time and is presented graphically in Figures 8 and 9. The results indicate that the temperature of each prototype increased steadily over time. It was observed that all prototypes exhibited a similar pattern with a significant temperature rise in the first 15 min. This test serves to evaluate the ability of the prototypes to reduce the rate of heat

transmission, thereby providing occupants with sufficient time to evacuate in the event of a fire. The observed phenomenon is attributed to the effective thermal insulation provided by vermiculite and perlite, which have porous structures that dissipate heat during a fire test. B1 showed a rapid and gradual increase in temperature after 15 min, possibly due to physical and chemical reactions of the intumescent binder. Subsequently, all prototypes exhibited a similar temperature profile, steadily increasing until the 120 min mark. Notably, B2 displayed the lowest temperature compared to the others, indicating that it offered the best fire-retardant performance and protection.

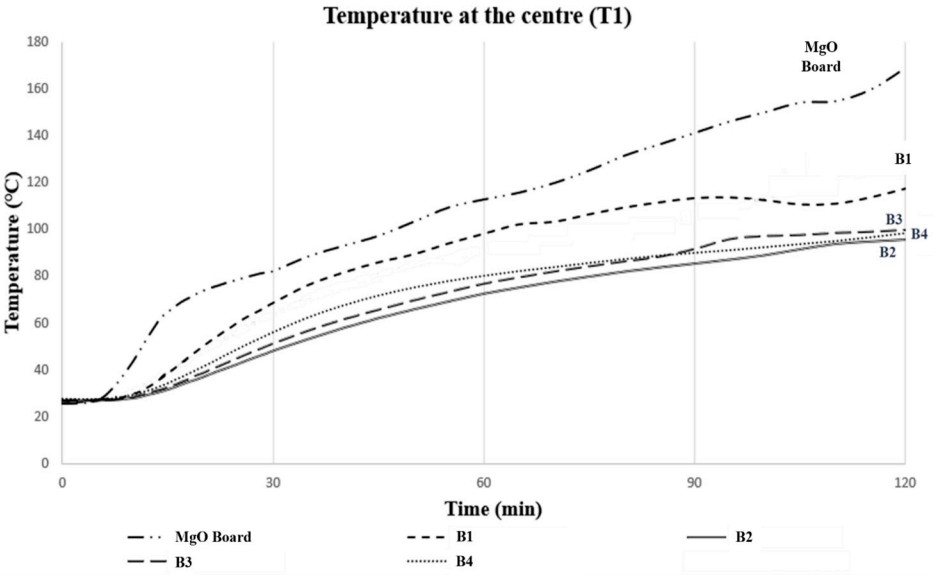

**Figure 8.** Evolution of temperature of fire-resistant boards at the center (T1).

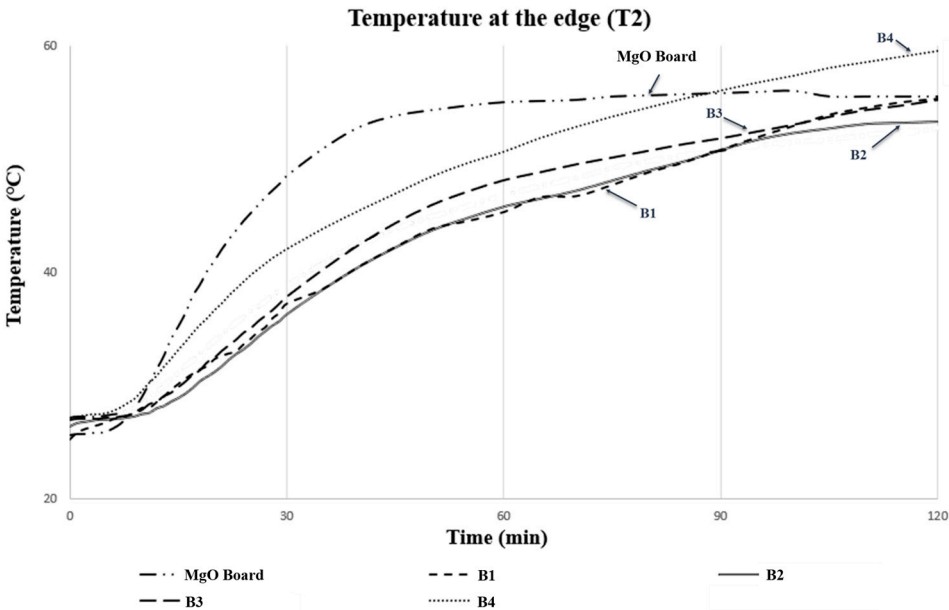

**Figure 9.** Evolution of temperature of fire-resistant boards at the edge (T2).

The comparison between prototype B2 and a commercial prototype regarding the temperature rise (measured at the center—T1 and edge—T2) is evaluated. The results showed that after 120 min of exposure to fire, prototype B2 had a lower temperature rise rating at T1 and T2, indicating a superior quality of low heat transmission compared to the commercial prototype.

An analysis of temperature increase at various points showed that prototype B2 exhibited superior fire retardancy characteristics as compared to the commercial prototype, due to its lower equilibrium temperature. The commercial prototype recorded the highest temperatures at T1 (165.6 °C) and T2 (54.0 °C), whereas prototype B2 demonstrated better fire retardancy properties with maximum temperatures recorded at T1 and T2 being only 92.6 °C and 51.5 °C, respectively. The difference in density between prototype B2 (density = 624.72 kg/m$^3$) and the commercial MgO fire-rated board prototype (density = 963.71 kg/m$^3$) may have played a role in this outcome. Additionally, the successful amalgamation of the intumescent flame-retardant binder with vermiculite and pearlite helped to reduce the rate of heat transmission, which resulted in a temperature decrease of up to 73.0 °C in prototype B2 during the 2 h fire test as compared to the commercial prototype.

To ensure the safety and reliability of fire-rated boards in buildings, it is crucial to conduct fire endurance and temperature rise tests. Therefore, in this study, both the commercial prototype and prototype B2 underwent such tests to evaluate their effectiveness in the event of a fire. The purpose of conducting the fire endurance test was to assess the ability of the fire-resistant board prototypes to withstand certain fire conditions for a specific period without any loss of integrity or significant leakage. One way to compare the effectiveness of the two prototypes was through a test assessing their ability to resist fire, which included evaluating their fire-resistance rating, heat transmission rate, and visual observations. For the purpose of comparison, a commercial fire-rated board prototype utilizing magnesium oxide board was used alongside sample B2. The results of the fire endurance test, temperature rise test, and smoke observation from the tested prototypes are obtained. According to the findings, both prototypes demonstrated the ability to withstand the specified fire condition for 2 h without any breach in their integrity or significant leakage, as shown in Figure 10. However, prototype B2 exhibited no smoke and only a slight odor, indicating that the materials used in its production are environmentally friendly. In contrast, the commercial prototype produced a dense white smoke with a strong odor during the test.

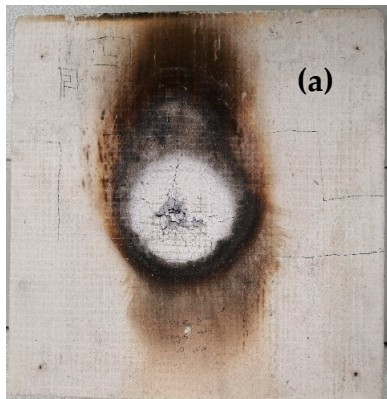 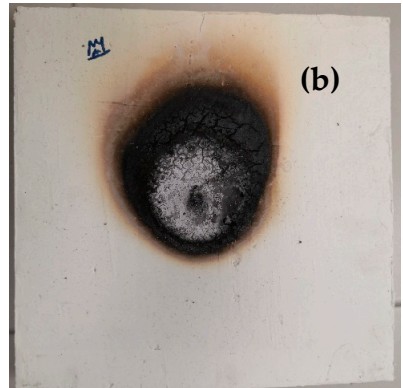

**Figure 10.** The boards after the 120 min fire test: (**a**) Commercial; (**b**) B2 fire-rated.

### 3.5. Three-Point Flexural Test

To evaluate the mechanical performance of the fire-resistant board prototypes, the flexural modulus and flexural strength (flexural stress) were measured in this test for five fire-rated boards as shown in Table 4. This was conducted as a component of the research project. Additionally, the flexural modulus and flexural strength of fire-resistant board prototypes are important mechanical properties that affect their overall durability and reliability. These properties can also determine whether the prototype can withstand heavy loads or high impacts, which are important factors for building safety. The flexural strength is particularly important for fire-resistant boards because they must be able to withstand not only the high temperatures of a fire but also the force of people during handling. The results

of this test can provide valuable insights into the mechanical properties of fire-resistant boards and help to improve their design and performance.

**Table 4.** Flexural stress and flexural modulus of all the prototype samples.

| Sample after the Test | Flexural Stress, σf (MPa) | Flexural Modulus, Ef (N/mm²) |
|---|---|---|
| B1 | 2.61 | 417 |
| B2 | 3.54 | 208 |
| B3 | 0.96 | 250 |
| B4 | 1.30 | 292 |
| MgO board | 2.80 | 313 |

As per the findings of Aghaee and his co-workers, the commercial prototype made of magnesium board exhibited low ductility and collapsed immediately after the occurrence of the first crack [37]. The flexural behavior of this prototype was entirely linear, indicating no plastic behavior. Hence, the prototype failed in a brittle manner as soon as it reached the peak load, breaking abruptly into two pieces. The outcomes of the three-point flexural test performed on all the prototype samples are illustrated in Table 4. It can be observed from the figure that prototypes 1 and 5 displayed a brittle failure behavior resembling that of the MgO board prototype. This implies that these prototypes may be susceptible to fracture or damage when subjected to high-impact forces. To avoid plagiarism, it is crucial to rephrase the sentences and use synonyms or alternate phrasing to convey the same information.

The second prototype demonstrated excellent ductility, as it was able to withstand higher flexural strength following the initial crack and exhibited higher values of flexural

strain. On the other hand, prototypes B3 and B4, which had lower flexural strength, displayed a progressive failure mode and exhibited greater ductility than B1 and MgO boards, although still lower than prototype B2.

Prototype B2 showed the highest flexural stress (3.54 MPa), indicating that it was able to withstand more stress impact. Additionally, prototype B2 had the lowest flexural modulus (208 N/mm$^2$), which means that it was more ductile than other prototypes. On the other hand, prototypes B1 and MgO boards had higher flexural modulus values (417 N/mm$^2$ and 313 N/mm$^2$, respectively), suggesting that they were very stiff and brittle, making them more prone to breaking when subjected to force. Prototypes B3 and B4 had weaker flexural properties than other prototypes but did not break into two pieces.

Prototype B2 exhibited elastic–plastic behavior and greater ductility, potentially due to the strong bonding between the VAC binder and the flame-retardant substances. This bonding enabled the material to span the cracked section, arrest crack propagation, and ultimately result in a ductile failure. In summary, the findings indicate that prototype 2 outperformed the other tested prototypes in terms of flexural properties.

*3.6. Acoustic Test*

3.6.1. Sound Absorption Measurement

The assessment of sound absorption is a frequently employed technique for gauging the sound absorption capability of materials. More specifically, the objective of this procedure is to establish the absorption coefficient of prototype samples at varying frequencies. The absorption coefficient is a numerical value ranging from 0 to 1, with 1 indicating that all sound is absorbed and 0 denoting that no sound is absorbed. This coefficient is used to express the level of sound absorption achieved by the material. According to Siemens Company in Malaysia, absorption ($\alpha$) for hardwood is approximately 0.2, i.e., $\alpha = 0.20$. In this research project, the fire-resistant board samples fall within the same group as the hardwood. Table 5 shows the absorption coefficient of all prototype samples for sound absorption measurement. It can be observed that all the sample prototypes have an absorption ($\alpha$) of approximately 0.2. Therefore, this has proved that the sample prototypes are still within the range of the acoustic standard for hardwood. However, absorption values are very subjective and may depend on the material properties themselves and environmental effects as well as the sample positions in the impedance tube during the test. This is to note that the softer the material, the higher the sound can be absorbed. The denser and harder the material, the lesser the chance of the sound being absorbed.

**Table 5.** Absorption coefficient of all prototype samples with frequencies.

| Sample Prototype | Frequency (Hz) | | | | | | Average AC |
|---|---|---|---|---|---|---|---|
| | 125 | 250 | 500 | 1000 | 2000 | 4000 | |
| | Absorption Coefficient (AC) | | | | | | |
| B1 | 0.03 | 0.22 | 0.20 | 0.03 | 0.12 | 0.23 | 0.14 |
| B2 | 0.24 | 0.22 | 0.19 | 0.03 | 0.30 | 0.21 | 0.20 |
| B3 | 0.19 | 0.25 | 0.20 | 0.05 | 0.20 | 0.11 | 0.17 |
| B4 | 0.12 | 0.25 | 0.20 | 0.06 | 0.06 | 0.43 | 0.19 |
| MgO board | 0.07 | 0.18 | 0.19 | 0.07 | 0.23 | 0.40 | 0.19 |

3.6.2. Transmission Loss Measurement

Sound transmission loss measurement is a technique to measure the sound energy quantification related to how the sound energy is prevented from travelling through an acoustic treatment. Hence, this test is to measure the transmission-loss performance of the prototype samples in different frequencies. The transmission loss represents how many decibels of acoustic absorber reduces the incident sound energy during the transmission

loss measurement. The higher the transmission loss in decibels, the lesser the sound transmitted will be.

Referring to the graph in Figure 11, it shows the sound transmission of all prototype samples for sound transmission loss measurement. All of them have indicated a similar pattern of transmission loss in the range of approximately 35 dB to 50 dB with different frequencies. It is well known that the transmission loss value of a barrier is heavily influenced by the frequency of the sound being transmitted. For instance, at 850 Hz (refer to the blue line in the graph), the acoustic absorber reduces the incident energy in prototypes B1, B2, B3, B4, and MgO by approximately 30 dB, 34 dB, 20.5 dB, 44 dB, and 37.5 dB, respectively. Sound transmission loss is a useful metric for evaluating the effectiveness of a barrier in reducing the transmission of sound energy. It should be noted, however, that the results of bench testing may differ slightly from those obtained onsite due to variations in environmental conditions.

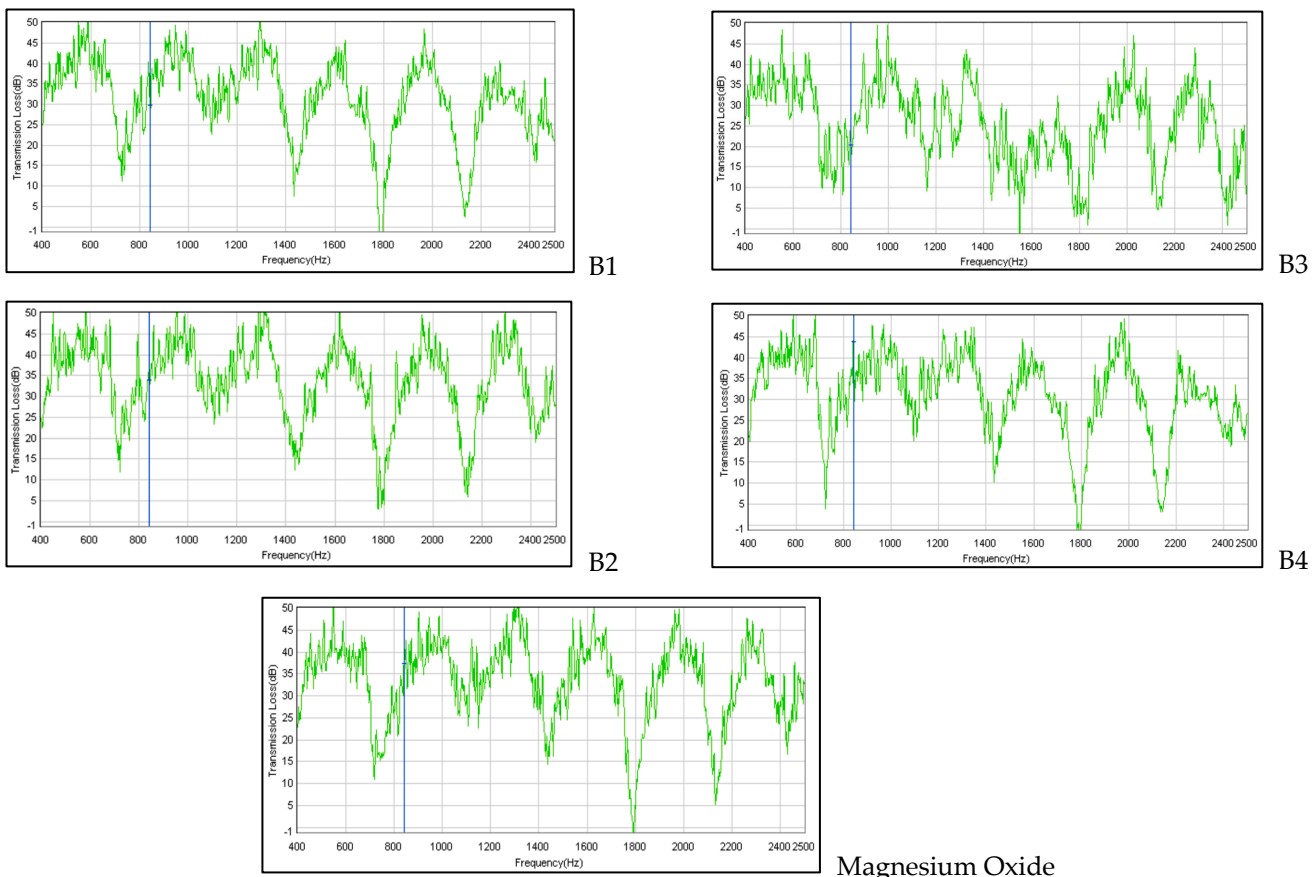

**Figure 11.** Sound transmission loss measurement of all prototype samples with various frequencies.

## 4. Conclusions

The experimental tests conducted in this study led to several noteworthy conclusions. The choice of appropriate combinations of fire-rated materials has a direct influence on the performances of fire protection and mechanical properties of the fire-resistant board. Out of all the intumescent binder samples tested, A2 was modified by adding aluminum hydroxide (3 wt.%) and renewable CES nano bio-filler (3 wt.%); it exhibited excellent fire-rated performance was enhanced in terms of mechanical, physical, and chemical properties. Prototype B2, which incorporated intumescent binder formulation A2, exhibited better test results than the currently available commercial fire-rated MgO board. In conclusion, the incorporation of a new and innovative intumescent binder in the fire-rated board has proven to be effective against a 2 h fire rating with good mechanical and acoustic properties.

**Author Contributions:** Conceptualization, M.C.Y. and M.K.Y.; Methodology, M.C.Y.; Software, M.K.Y.; Validation, M.C.Y. and M.K.Y.; Formal Analysis, M.C.Y. and R.K.K.Y.; Investigation, M.C.Y.; Resources, M.C.Y.; Data Curation, M.K.Y. and R.K.K.Y.; Writing—Original Draft Preparation, M.C.Y.; Writing—Review and Editing, M.C.Y., M.K.Y. and R.K.K.Y.; Visualization, M.K.Y. and R.K.K.Y.; Supervision, M.C.Y.; Project Administration, M.C.Y. and M.K.Y.; Funding Acquisition, M.C.Y. All authors have read and agreed to the published version of the manuscript.

**Funding:** This project was funded by the University of Tunku Abdul Rahman Research Fund (UTARRF) with project number IPSR/RMC/UTARRF/2021-C1/Y01 and the Fundamental Research Grant Scheme (FRGS) under the project number FRGS/1/2022/TK088/UTAR/02/31.

**Institutional Review Board Statement:** Not applicable.

**Informed Consent Statement:** Not applicable.

**Data Availability Statement:** https://www.mdpi.com/ethics.

**Acknowledgments:** The authors would like to express their sincere gratitude to the University of Tunku Abdul Rahman Research Fund (IPSR/RMC/UTARRF/2021-C1/Y01) and the Fundamental Research Grant Scheme (FRGS/1/2022/TK088/UTAR/02/31).

**Conflicts of Interest:** The authors declare no conflict of interest.

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
