# Peer review of "Experimental Analysis of Lightweight Fire-Rated Board on Fire Resistance, Mechanical, and Acoustic Properties"

_fire, doi:10.3390/fire6060221_

Round 1
Reviewer 1 Report
The work presented in this manuscript dealing with interesting topic and fire safety of building materials is mandatory requirements , however, there some points need to be addressed as listed.
1. Title of manuscript need to modified reflecting the content of mansucript as this is not review to give broad title.
2. English need revision to address typos and some mistakes.
3. Introduction section need enriched with other commercial flame retardants materials for building materials and I recommend citation of this reports
Materials Chemistry and Physics 168 (2015) 147-158
Construction and building Materials 201 (2019) 138-148
4. Authors need to do single burning item test for this material if possible.
5. deeper discussion about role of each flame retardant filler is required.
6. What is the role of TiO2 in the flame retardation action ?
English need revision to address typos and some mistakes throughout the manuscript.
Reviewer 2 Report
After reading the text, in general, I assess the scientific quality of the publication as very well.
The authors presented the results of research of chosen parameters lightweight firerated board (LFRB)
The manuscript sent for evaluation consists of 20 pages, including 17 pages of text with tables and figures, and 3 pages for references (35). The main author has 4 auto citations.
Abstract would be to contain all the necessary information, such as methods, results, main statements. Please, add aim of abstract.
The visual documentation is illustrative and clear.
the article is very rich in content. The Results and Discussion chapter describes the obtained experimental data.
I ask the authors to supplement the discussion section by comparing the results with other authors.
Reviewer 3 Report
This manuscript presents a study on the fire resistance, acoustic, oxygen/carbon ratio and mechanical properties of the lightweight fire-rated board. The manuscript could be accepted if the following problems/ questions can be addressed:
1.The literature review should introduce the concepts of active fire prevention and passive fire prevention, so that readers can have a clearer understanding of their differences.
2. The literature review lacks previous research on IB and IM.
3.In section 2.2, please explain whether the formula mentioned in impedance tube test has theoretical basis.
4. What does α in Formula 2.3 mean? There is no explanation in the manuscript.
5.In the furnace test in Section 3.2, an additional graph can be added to compare and analyze the carbon layer thickness of the four samples A1-A4 at 500 ℃ and 600 ℃ respectively.
6. In line 355, why do the authors write "B4 showed a rapid and gradual increase in temperature after 15 minutes"? The temperature change of B1 seems to be more obvious.
7. In this paper, there are only some diagrams of the instruments, but there are no diagrams of the test process.
8. The conclusion section provides less summary of the experimental results, and it is recommended to summarize the conclusion in a clear and organized way.
Round 2
Reviewer 3 Report
The authors have revised the manuscript accoring to the suggestions. Hence, the manuscript can be published in current form.